# Bioaccessibility and Cellular Uptake of Lutein, Zeaxanthin and Ferulic Acid from Muffins and Breads Made from Hairless Canary Seed, Wheat and Corn Blends

**DOI:** 10.3390/foods12061307

**Published:** 2023-03-19

**Authors:** El-Sayed M. Abdel-Aal, Iwona Rabalski, Christine Carey, Tamer H. Gamel

**Affiliations:** Agriculture and Agri-Food Canada, Guelph Research and Development Centre, 93 Stone Road West, Guelph, ON N1G 5C9, Canada

**Keywords:** hairless canary seed, carotenoids, phenolic acids, in vitro bioavailability, baked products, digesta cleanup

## Abstract

Using a simulated gastrointestinal digestion model combined with a Caco-2 cell model, this study aims to assess the bioaccessibility and cellular uptake of dietary lutein, zeaxanthin, and ferulic acid from muffins and bread prepared from blends of hairless canary seed (HCS), wheat, and corn. Residual digestive enzymes damaged the Caco-2 monolayer and necessitated the requirements for the additional clean-up of the digesta. Several digesta cleanup treatments were examined, and the C18 column, along with AEBSF inhibitor, was selected as the most effective treatment. However, the cleanup treatment reduced lutein, zeaxanthin, and ferulic acid concentrations. The bioaccessibility of lutein from muffins was high at 92–94% (without clean-up) and 81–86% (with cleanup); however, the cellular uptake was low (7–9%). The bioaccessibility and cellular uptake (4–11%) of zeaxanthin were lower than lutein. Ferulic acid from muffins exhibited a wide range of bioaccessibility for non-cleanup (105–229%) and clean-up (53–133%) digesta samples; however, cellular uptake was very low (0.5–1.8%). Bread made from wheat/HCS had higher lutein bioaccessibility (47–80%) than the control bread (42%), with an apical cellular uptake ranging from 4.3 to 9.2%. Similar to muffins, the bioaccessibility of zeaxanthin from bread was lower than lutein, while ferulic acid had a fairly high bioaccessibility at 98–103% (without clean-up) and 81–102% (with cleanup); however, zeaxanthin cellular uptake was low (0.2%). These results suggest that muffins and bread could boost the daily consumption of lutein, zeaxanthin, and ferulic acid, allowing for a small portion to be absorbed in the small intestine.

## 1. Introduction

Wholegrain foods are considered good sources of bioactive compounds, such as carotenoids and polyphenols. In this regard, lutein and zeaxanthin are the principal carotenoids in corn and hairless canary seed (HCS) [1], while ferulic acid is the most abundant non-flavonoid polyphenol in cereal grains [2]. The role of lutein and zeaxanthin in the health of eyes and cognitive function [3,4] has been well evidenced. Likewise, ferulic acid plays significant roles in human health in the area of oxidative stress, diabetes, cancer, and cardiovascular disease [5,6]. The hairless canary seed (*Phalaris canariensis* L.) is a member of the grass family (*Gramineae*), similar to wheat and corn. It was developed through a successful breeding program to eliminate potential health threats associated with the presence of tiny siliceous hairs found in the traditional hairy variety [7]. The HCS was subsequently granted food approval by Health Canada [8] and received a generally recognized as safe (GRAS) status by the US Food and Drug Administration [9]. The grain possesses a unique nutrient composition with relatively high levels of protein (22.7–23.7%) and oil (7.3–7.7%), along with comparable amounts of starch (55.8–57.2%) [10]. Due to its high content of carotenoids, phenolic acids [11], and bioactive peptides [12,13], HCS represents a promising functional food ingredient. The potential use of HCS as a proper ingredient for the gluten-free foods [14] and wholegrain healthy products [11,15,16,17] can widen commercial applications and consumption of these compounds through diet.

To better understand what happens to food after ingestion, foods or meals are subjected to in vitro digestion (salvia, gastric and intestinal) and conditions that simulate the human digestive system and render the food nutrients accessible. Coupled with an in vitro Caco-2 cell culture model, the uptake and transport (e.g., in vitro bioavailability) of nutrients and bioactive compounds [18,19] can be evaluated. To date, several models exist, including the dynamic gastrointestinal model (e.g., TIM™), which is a sophisticated and expensive system whereby many parameters of the human digestive system are simulated, while the Caco-2 cell model is widely used to assess nutrient uptake and bioavailability in vitro. Garrett et al. [20,21] previously evaluated the relative bioavailability of carotenoid-rich baby food meals prepared from vegetables (carrot, spinach, and tomato) and chicken or meat and demonstrated that the in vitro bioavailability of lutein, lycopene, α-carotene, and β-carotene is concentration and time-dependent. As such, the model was determined to be a useful tool in assessing the uptake of carotenoids and other food components. In our previous study, lutein concentration in the aqueous phase of digesta was the most important factor in determining cellular lutein uptake from muffins, cookies, and flatbreads enriched with lutein and measured under fed or fasted-state digestion conditions [22]. Our study also demonstrated that cookies and muffins exhibited a higher lutein uptake than flatbreads (*p* < 0.05) due to their higher fat content, and further, lutein bioaccessibility was higher in the fed state, compared with the fasted state (*p* < 0.001). The simulated gastrointestinal digestion model has also measured the bioaccessibility of polyphenols from wholegrain wheat pasta [23] and commercial wheat pasta when fortified with chia seed by-products [24], whereby only a small proportion of free polyphenols were bioaccessible for absorption into the small intestine.

Recently hairless canary seed was used alone or in blends with corn or wheat to develop muffin and bread prototypes as potential healthy foods to boost the daily consumption of lutein, zeaxanthin, and phenolic acids [1,25], as their abundance in the human body was dependent on dietary intake. In those studies, the nutritional quality and overall acceptability of muffin and bread products were demonstrated, along with the impact of baking on carotenoids and phenolic acids. To further our knowledge of the dietary benefit of hairless canary seeds and estimate their potential absorption in the small intestine, the current study investigated the bioaccessibility and cellular uptake of lutein, zeaxanthin, and ferulic acid from muffins and bread made from blends of HCS, wheat, and corn.

## 2. Materials and Methods

### 2.1. Chemicals

HPLC grade methanol, ethanol, hexane, acetonitrile and methyl tert-butyl ether, and sodium hydroxide and hydrochloric acid were purchased from Fisher Scientific (Mississauga, ON, Canada), and Suprapur formic acid from VWR (Mississauga, ON, Canada). High-purity lutein 00012453 and zeaxanthin 00026504 standards were purchased from ChromaDex (Irvine, CA, USA). Ferulic acid was purchased from Sigma (Sigma-Aldrich Canada Ltd., Oakville, ON, Canada). Digestive enzymes, including α-amylase from human saliva A0521-2.5KU, pepsin P7000, and pancreatin 1760-25G, were purchased from Sigma Aldrich. Porcine bile extract SC214601 was purchased from Santa Cruz Biotechnology Inc. (Dallas, TX, USA). The enzyme inhibitors AEBSF [4-(2-aminoethyl)benzenesulfonyl fluoride hydrochloride] and PMSF (phenylmethylsulfonyl fluoride) were purchased from Millipore Sigma (Oakville, ON, Canada) and VWR Canada, respectively. The nano pure water was obtained from the Milli-Q integral water purification system (Millipore Ltd., Etobicoke, ON, Canada).

### 2.2. Grains and Flours

The hairless canary seed (yellow variety) was kindly provided by the Canary Seed Development Commission of Saskatchewan (CSDCS, Saskatoon, SK, Canada), and hard red spring wheat was obtained from the University of Saskatchewan (Saskatoon, SK, Canada). Yellow corn flour was purchased from the local market in Guelph, ON, Canada. The HCS grains were milled using the UDY cyclone mill (310-014, UDY Corporation, Fort Collins, CO, USA) into wholegrain flour, while the wheat grains were milled on the Brabender Junior mill into wholegrain flour (Quadrumat Junior, Brabender Instruments Inc., Duisburg, Germany).

### 2.3. Preparation of Bread and Muffin Products

Three low-fat muffins were prepared from HCS flour alone or in a blend with corn flour at a ratio of 1:1 and 1:2 (*w*/*w*). The muffin formulas and baking process are described in our previous study [1]. In brief, the muffin formula contained 120 g HCS flour or flour mix, 36 g sugar, 1.5 g salt, 3 g baking powder, 2.5 g egg white powder, and 5 g canola oil, in addition to 80–85 g of water. Three bread products were made from the composite flour of wheat and HCS blends at a ratio of 85/15, 75/25, and 50/50 (*w*/*w*), in addition to a control bread (100% wheat flour). The bread ingredients and baking conditions are reported in our previous study [25], where the sourdough bread was prepared using 75 g flour, 50 g sour, 2 g salt, 5 g sugar, 3 g corn oil, 0.5 g dry yeast, and 37.5 g water. All baking trials were, at a minimum, in triplicate, and fresh samples were taken for in vitro digestion experiments.

### 2.4. In Vitro Simulation of Gastrointestinal Digestion

Simulating the 3 steps of human digestion, our study followed the in vitro digestion procedure according to the methodology described by Pigni and others [24]. The oral digest involved homogenizing 2 g of bread or muffin for 30 s at 12,000 rpm in 2 mL of an α-amylase solution containing 75 U/mL using Polytron PT10/35 (Kinematica AG, Lucerne, Switzerland). Nano-pure water (2 mL) was added, and the content was adjusted to pH 2.0 with 1 N HCl. For gastric digestion, 0.5 mL of pepsin solution (20 mg of pepsin dissolved in 0.1 N HCl) was added to each tube, which was then placed in a 37 °C shaking water bath SWB25 (Thermo Haake, Newington, NH, USA) for 2 h. The intestinal digestion was carried out by adding 3 mL of pancreatin/porcine bile solution (15 mg of pancreatin enzymes and 75 mg of porcine bile in 0.1 M NaHCO_3_), which was then pH-adjusted to 7.5 with 2 N NaOH and incubated in a 37 °C shaking water bath for 3 h. Following the 3-step digestion, 80 µL of a 100 mM AEBSF enzyme inhibitor solution was added, and the digested materials were vortexed and centrifuged at 4400 rpm for 10 min. The aqueous phase of the digesta was aliquoted to a clean 15 mL tube and stored at −80 °C until further cleanup using the SPE C18 SampliQ column, as described below. Bioaccessibility is defined as the amount transferred to the aqueous phase (micellarization) and is presented as the % of the original concentration in the muffin or bread.

### 2.5. Cleanup Treatments

In preliminary experiments, the digestion product damaged the Caco-2 monolayer. This is consistent with our previous study [22], whereby high bile and pancreatin concentrations under physiologically representative conditions were cytotoxic to the Caco-2 monolayer. Consequently, several cleanup treatments were evaluated to ensure Caco-2 cell viability and the integrity of lutein and ferulic acid as well. Following the 3-step digestion, clean-up treatments were performed and immediately added to the cell culture. The removal of enzymes and other harmful substances present in the intestinal digests by physical means such as dialysis using an Amicon Ultra -15 cellulose membrane with a molecular weight cut off 10K (MiliporeSigma), SPE columns: Amberlite XAD7 and IR120 (used for moderately polar retention; MilliporeSigma), Waters Oasis HLB (used for polar/nonpolar retention; Waters Canada Ltd., Mississauga, ON, Canada) and Agilent Alumina-N (used for polar retention) or Agilent C18 SampliqQ (used for non-polar retention; Agilent Technologies Canada, Mississauga, ON, Canada) were individually evaluated. All columns were sequentially hydrated three times with 2 mL of nano-pure water and 2 mL of 95% ethanol followed by 2 mL of nano-pure water. Then, 2 mL of the digesta aqueous phase or supernatant was loaded onto the column, with approximately 2 mL collected as fraction 1, and the retained compounds were eluted with 95% ethanol to obtain fraction 2 (nearly 2–2.2 mL). Each fraction was adjusted to 2 mL using a nitrogen evaporator N-EVAPTM111 (Organomation Associates Inc., Berlin, MA, USA) and examined for the presence of carotenoids and phenolic acids, and Caco-2 cell viability was monitored on a Vista Vision microscope. Carotenoids and phenolic acids were present in fraction 1 of all the columns, prompting further investigation. Other treatments that were evaluated include the adjustment of digesta to pH 2.0 (1 N HCl) or adding AEBSF or PMSF to inactivate pancreatic enzymes in the digesta. Additionally, cooling in an ice box or liquid nitrogen to stop the action of enzymes was also evaluated. Following each treatment, the digesta samples were double-filtered (as mentioned below), and 1 mL of the test medium containing the aqueous fraction was diluted 1:3 (*v*/*v*) with DMEM medium. Following a 4 h incubation, the viability of the Caco-2 monolayer was reported as either monolayer intact (MI) or monolayer detached (MD). Lutein and ferulic acid were also determined following the experimental digesta clean-up.

An evaluation of all digesta clean-up treatments determined the optimal clean-up procedure to include the addition of AEBSF, followed by the C18 SPE SampliQ column. The digesta (−80 °C) was defrosted at room temperature, vortexed, and then, three aliquots (2 mL each) were loaded on three separate columns. Approximately 2.1–2.2 mL of fraction 1 was collected from each column. The volume was adjusted to 2 mL using a nitrogen evaporator. The three fractions were pooled to give a total volume of 6 mL, filtered in a syringe through a 0.45 µm Nylon filter (Chromatographic Specialties CS25N452, Brockville, ON, Canada), followed by an additional filtration through the 0.2 µm GHP filter (Pall 4554T, Pall Corporation, Port Washington, NY, USA). Aliquots of 100 µL of the cleaned-up aqueous phase were saved for carotenoids and phenolic acids analyses, and the remaining 5.9 mL was used for cell culture experiments.

### 2.6. Cellular Uptake of Lutein, Zeaxanthin and Ferulic Acid Using Caco-2 Cell Model

The Caco-2 cell line HTB-37 was purchased from Cedarlane Laboratories (Burlington, ON, Canada): a distributor of American Type Culture Collection (Rockville, MD, USA). The cells were maintained in high glucose Dulbecco’s modified Eagle medium supplemented with 10% (*v*/*v*) heat-inactivated fetal bovine serum, non-essential amino acids (10 mL/L), 1% antibiotic-antimycotic solution, sodium pyruvate (1 mmol/L), and HEPES (15 mmol/L). Unless otherwise indicated, all cell culture reagents were obtained from Gibco^TM^, Grand Island, NY, USA. The cells were grown at 37 °C in a humidity controlled 5% CO_2_ incubator, with a medium that could be changed every other day. Caco-2 cells were sub-cultured at about 70–80% confluency with 0.25% trypsin-EDTA and seeded in a cell culture flask of a 75-cm^2^ area (Corning Incorporated, Lowell, MA, USA) or 12-well flat bottom tissue culture plates (Corning, NY, USA) at a density of 1 × 10^4^ cells/cm^2^. Caco-2 cultures were between passages 42 and 58 and used for experiments between 11 and 14 days post confluency since, at that time, the Caco-2 cells exhibited maximal differentiation [26]. Cell viability was assessed by trypan blue staining, comparing the test sample and control well with an equivalent volume of 0.9% saline. Microscopically, the wells appeared similar, and cell viability by trypan blue stain was >96%. Monolayers were washed twice with 1 mL of Hank’s Balanced Salt Solution (HBSS, Gibco^TM^) prior to adding 1 mL of the test medium containing 0.75 mL complete DMEM and 0.25 mL of the aqueous phase of the in vitro digesta or 0.25 mL saline (control) to triplicate wells. Following a 4 h incubation at 37 °C and 5% CO_2_, the medium was removed, and the monolayers were washed twice with HBSS containing 5 mmol/L sodium taurocholate to remove carotenoids adhering to the cell surfaces [21]. Ethanol (1 mL, 95%) was added to each well, and samples were stored at −80 °C until HPLC or UPLC analysis. Each sample was added in triplicate to the monolayer. The negative and positive controls were also included for each trial. Cellular uptake was determined as the amount of the bioactive compound retained by Caco-2 cells following a 4 h incubation with sample digesta and is presented as the % of the original concentration in muffin or bread. For comparison purposes, the data are also presented in absolute amounts.

### 2.7. Analysis of Carotenoids and Phenolic Acids

Analyses of carotenoids and phenolic acids were carried out on muffins, bread, intestinal digesta, and Caco-2 cell samples using HPLC and UPLC, respectively. The extraction of carotenoids and phenolic acids from muffin and bread products was performed according to our previously published procedures [1,25]. Samples were filtered through a 0.45 µm nylon filter and 0.2 µm GHP filter and used for the analyses. To evaluate the carotenoid and phenolic acid cellular uptake, an additional 1mL of 95% ethanol was added to harvest Caco-2 cells using a rubber scrapper, which was then transferred to a graduated glass tube, vortexed for 30 s, sonicated for 1 min, and vortexed again for 30 s. The tube was placed under a nitrogen stream to evaporate the ethanol to near dryness before it was then redissolved in 0.3 mL of water-saturated butanol for carotenoid analysis or 0.3 mL nano pure water for phenolic compounds analysis. The tube contents were sonicated for 3 min, filtered using a 0.2 µm GHP syringe filter, and subsequently analyzed by HPLC or UPLC. Carotenoids were separated and quantified using the YMC C30 column (15 cm × 4.6 mm × 3 µm) (Waters Canada Ldt., Mississauga, ON, Canada), while phenolic acids used the Agilent SB C18 column (10 cm × 2.1 mm × 1.8 µm) (Agilent Technologies Canada, Mississauga, ON, Canada) as described previously [1,25]. The amount of lutein presented in this study is the sum of all lutein isomers, including trans-lutein and a possible co-eluted cis-lutein.

### 2.8. Statistical Analysis

Data were statistically analysed using Sigma-Plot version 14.5 (Systat Software Inc., San Jose, CA, USA). One-way ANOVA was used to determine differences among muffin and bread products using the Tukey method and considered significant at *p* < 0.05.

## 3. Results and Discussion

### 3.1. Treatment of Intestinal Digests

Although lutein and ferulic acid were found in the digested muffin and bread aqueous fractions following the simulated gastrointestinal digestion, the untreated in vitro intestinal digests (as is) damaged the Caco-2 cells, and the monolayer remained detached at the end of a 4 h incubation period (Table 1). It has been reported that the high concentration of pancreatic enzymes and bile salts used in the simulated gastrointestinal digestion under the fed state conditions could damage Caco-2 cells [22]. Our results indicate that although lutein and ferulic acid were accessible and could potentially be absorbed in the small intestine, additional clean-up treatments were needed to eliminate the negative effect of pancreatic enzymes and other harmful components on Caco-2 cells. Table 1 summarizes several chromatographic, physical, and chemical treatments that were evaluated for their impact on the Caco-2 cell viability and integrity of lutein and ferulic acid. Although intestinal digests adjusted to pH 2.0 deactivated the enzymes and the monolayer remained intact over the incubation period (4 h), the pH treatment negatively affected lutein, as it was not detected by HPLC in the cleanup of the digesta samples. As a result, the pH treatment was not considered an appropriate method for assessing the lutein uptake in vitro. To support this conclusion, lutein has previously been reported to be unstable against an extreme pH [27,28] and, hence, this treatment is not recommended for the measurement of the cellular uptake of carotenoids due to the potential oxidization and/or degradation of lutein. Although acidification at pH 2.0 and heating at 100 °C for 4 min have been suggested to prevent the enzymatic degradation of Caco-2 cells [18], these treatments would be more appropriate for assessing the cellular uptake of invulnerable nutrients, such as minerals. It is worth mentioning here that although the clean-up digesta treatments do not physiologically represent in vivo conditions, these treatments are necessary to maintain the survivability of Caco-2 cells under the fed state condition. Equally as important is the treatment needed to preserve the integrity of lutein and other vulnerable molecules. The use of the in vitro co-culture of Caco-2 cells and HT29-MTX cells (a human mucus-producing cell line) can generate a protective mucus layer against digestive enzymes, as well as represent a more physiological and realistic approach to in vivo conditions [18,29]. In this respect, additional research is warranted to evaluate the effectiveness of the HT29-MTX mucus layer in preserving Caco-2 cell viability in simulated gastrointestinal digestion models to ensure that the mucus layer does not impede the uptake of bioactive compounds.

Cooling the intestinal digests in an ice box or freezing in liquid nitrogen was insufficient to inactivate the enzymes, as indicated by the monolayer detachment (Table 1). Although the dialysis of intestinal digests through an Amicon Ultra-15 membrane maintained Caco-2 cell viability, lutein was retained on the membrane (yellow coloration observed) and not detected in the cleanup digesta. As the 10 K membrane cut-off is much higher than the molecular weight of lutein, carotenoids should easily pass through; however, our results suggest that lutein could be retained and/or attached to large molecules (e.g., soluble protein and/or soluble fiber) that are present in the digesta during the dialysis treatment. It has been reported that lutein, zeaxanthin, and β-carotene bind to the protein, and the protein–carotenoid complex can interact with lipid membranes [30].

The use of SPE columns or cartridges with Amberlite XAD7 or IR120 was ineffective in removing enzymes, resulting in the cell monolayer becoming detached (Table 1). Similar results were also found with the SPE column Oasis HLP, as the monolayer was also damaged at the end of the 4 h incubation period. In contrast, the cleanup of the intestinal digests with an SPE column Alumina-N or C18 SampliqQ effectively removed enzymes and other potentially harmful components, allowing the monolayer of the Caco-2 cells to remain intact over the 4 h incubation period. While both columns maintained monolayer health, the C18 SampliqQ column was selected for subsequent experiments as the column exhibited slightly better lutein recovery (about 10–15% higher based on the UPLC area). The addition of AEBSF or PMSF protease inhibitors prior to the C18 SampliqQ column cleanup further improved monolayer attachment. Having lower toxicity and better solubility in water, AEBSF was selected to be used in combination with the C18 SampliqQ column. As noted in Table 1, clean-up with only a protease inhibitor was insufficient to maintain an intact monolayer over the incubation period. The effect of pure lutein (10–115 µg/mL) and ferulic acid (42–84 µg/mL) on the health of Caco-2 cells was also evaluated. A high lutein concentration (115 µg/mL) formed some voids in the monolayer at the end of the 4 h incubation period, suggesting that high lutein concentrations in food or meals could affect the monolayer integrity at a longer incubation time. Based on the above results, AEBSF inhibitor and C18 column were used in combination to clean up intestinal digests for the assessment of the cellular uptake of lutein, zeaxanthin, and ferulic acid from muffins and bread.

### 3.2. Bioaccessibility and Cellular Uptake of Carotenoids and Ferulic Acid from Muffins

In our previous study [1], three low-fat muffins made from hairless canary seed (HCS) alone or in blends with corn were developed and evaluated in terms of carotenoid composition and the impact of baking on carotenoid content. In comparison to wheat muffins, HCS- and corn-based muffins are relatively rich in lutein and zeaxanthin. The concentration of lutein and zeaxanthin in the muffins showed significant differences among the three muffin formulations (Table 2). The addition of corn resulted in a significant increase in lutein and zeaxanthin, particularly for the HCS/corn muffin at a 1:2 ratio. This is associated with the reduced risk of age-related macular degeneration and cataracts, in addition to their positive functions in vision and cognition [3,4]. Since lutein, and zeaxanthin cannot be synthesized by the human body and, therefore, must be provided in food formulations. In addition to the unbound or free lutein, HCS is also rich in lutein mono- and di-esters, such as lutein 3-O-linoleate, lutein 3-O-oleate, and lutein di-linoleate, while corn is rich in free zeaxanthin [1]. Other carotenoids are present in HCS and corn having a total content of 7.6 and 12.9 μg/g, respectively [1].

As ferulic is the dominant phenolic acid in cereal grain products, muffins made from HCS and corn could represent a reasonable source of ferulic acid. The HCS muffin showed significantly higher concentrations of soluble ferulic acid than HCS/corn muffins (Table 2). The addition of corn significantly reduced the ferulic acid content in HCS/corn muffins, compared to 100% HCS muffins. In a previous study, ferulic and *p*-coumaric acids have been reported to be the dominant phenolic acids, with the bound form being primarily higher in HCS muffins compared with HCS/corn muffins and vice versa for the soluble phenolic acid fraction [1]. The baking process could liberate a portion of the bound phenolic acid fraction, resulting in an increase in the soluble phenolic acid fraction in muffins. The releasing rate of soluble ferulic acid during thermal processing could be different and dependent on the cereal type and processing conditions. In general, the increase in soluble or free phenolic acids could result in more bioavailable ferulic acid from muffin products.

When muffin products were subjected to the in vitro digestion protocol, the concentration of lutein, zeaxanthin, and ferulic acid changed compared to initial concentrations (Table 2). A slight decrease in lutein and zeaxanthin concentrations was observed, while ferulic acid had a wider range of increase (4–152%) which could possibly be due to the release of bound ferulic acid during the in vitro digestion process. The amount of lutein, zeaxanthin, and ferulic acid in the aqueous phase of the intestinal digests of muffin products (i.e., the bioaccessible fraction) was used to estimate the bioaccessibility of the three bioactive components (Figure 1). Despite a 92–94% bioaccessibility of lutein among the three muffins (no significant differences), the required digesta clean-up reduced lutein bioaccessibility to 81–86%, with no significant differences among the three (Figure 1A). The high-fat content in muffins has been shown to improve lutein stability in the micelles and thus its bioaccessibility [22]. Further, the study demonstrated that the bioaccessibility of lutein was dependent on food type, lutein concentration, fat content, and digestion protocol, i.e., fed versus fast state conditions [22]. Specifically, lutein bioaccessibility of muffins made from einkorn wheat or einkorn fortified with lutein was in the range of 37–54% subject to lutein concentration. In a separate study, the lutein bioaccessibility of five semolina pasta ranged from 63 to 78%, compared to 56–59% for egg pasta (n = 3), with no significant differences among the pasta products or between different stereoisomers of lutein [31]. Despite the high bioaccessibility of lutein, the cellular uptake of lutein in our study was 7–9% (Figure 1A). The observed decrease could be attributed to oxidative reactions and the low stability of micellar carotenoids during the 4 h incubation time with Caco-2 cells. It has been reported that the in vivo bioavailability of lutein is low and dose-dependent in healthy individuals when fed lutein-fortified fermented milk [32].

In contrast, significant differences in zeaxanthin bioaccessibility were observed among the three muffin products and could likely be attributed to the increased zeaxanthin concentration in corn-based formulations (Figure 1B). Although no significant differences were observed after the cleanup treatment, the bioaccessibility of zeaxanthin was lower than lutein, ranging from 63 to 79% for the untreated digesta samples and 38–48% for the cleaned-up samples. Similar to lutein, a portion of zeaxanthin was lost during the cleanup treatment. In a study, zeaxanthin bioaccessibility ranged from 63 to 88% in pasta without eggs and 48–59% in egg pasta, with no significant differences among the pasta products [31]. Consistent with other bioaccessibility studies, our results support the role of the high micellarization of lutein and zeaxanthin during the in vitro human digestive system. In other words, lutein and zeaxanthin in the aqueous micellar phase would be accessible in the small intestine. It has been reported that xanthophylls (e.g., lutein, zeaxanthin, and β-cryptoxanthin) are more easily released from food matrices and more efficiently micellarized than carotenes (e.g., lycopene, β-carotene, and α-carotene); however, carotenes are more efficiently taken up by the enterocytes [33]. Similar to lutein, the apical uptake of zeaxanthin was low (4–11%), with no significant differences among muffins (Figure 1B). In general, despite the high accumulation of lutein and zeaxanthin in the micelles during in vitro digestion, uptake by Caco-2 cells was very low. The absorption mechanism of carotenoids by enterocytes is still uncertain, and more than one mechanism could be involved, e.g., passive diffusion and receptor-mediated transport. In general, HCS/corn muffins could boost the daily consumption of lutein and zeaxanthin.

Due to the significant differences among muffin products in their content of ferulic acid (Table 2), significant variations in bioaccessibility were also found, both with or without cleanup (Figure 1C). Specifically, ferulic acid bioaccessibility prior to clean-up ranged from 105 to 229%, whereas the cleaned-up digesta samples had a bioaccessibility of 53–133%. Samples with a bioaccessibility of more than 100% were likely due to the release of a small portion of bound ferulic acid during the in vitro gastrointestinal digestion, as shown in Table 2. The observed significant loss of ferulic acid during the clean-up treatment could be related to interactions with other food molecules during the purification process. Individual phenolic acids in wheat have been found to have higher bioaccessibility after in vitro digestion (58–82%) than after in vitro colonic fermentation (22–48%) [34]. Similar to bioaccessibility, significant differences in the cellular uptake of ferulic acid were observed among the three muffin products and ranged from 0.5 to 1.8% (Figure 1C). This indicates that a small proportion of free ferulic acid could be accessible for absorption in the small intestine, as previously found in pasta products [23,24].

### 3.3. Bioaccessibility and Cellular Uptake of Carotenoids and Ferulic Acid from Breads

Bread is a major component of the human diet in many parts of the world and can be a suitable vehicle for many nutrients and health-enhancing components through the enrichment of wheat flour with ingredients that are rich in bioactive compounds. The current study used HCS as a source of bioactive peptides, minerals, vitamins, carotenoids, and polyphenols to replace a portion of wheat flour in bread recipes, with the intention of boosting the daily intake of carotenoids and phenolic acids. Replacing a portion of wheat flour with HCS resulted in an increase in the content of lutein, zeaxanthin, and soluble ferulic acid in bread (Table 3). Significant differences were found in these bioactive compounds between the control bread and wheat/HCS bread. Although the number of carotenoids and free ferulic acid in wheat/HCS bread was smaller than the suggested daily requirements, wheat/HCS bread can still boost the daily consumption of these healthy components [1,25]. In addition to carotenoids and ferulic acid, HCS is a recognized source of bioactive peptides [12,13], while wheat is a good source of dietary fiber. These compositional differences could make both grains complement each other in terms of health-enhancing attributes.

The concentrations of lutein, zeaxanthin and ferulic acid in the in vitro digested products were significantly different among the bread (Table 3). A portion of carotenoids was lost during the digestion process, but there was a slight increase in ferulic acid, which was probably due to the release of a small portion of bound ferulic acid during the in vitro digestion process. As indicated in Table 3 and consistent with results in muffin products, the cleanup treatment further reduced the number of carotenoids and ferulic acid. Since lutein, zeaxanthin, and ferulic acid are small polar molecules, the compounds could bind to proteins and/or polysaccharides that are present in the digesta. Lutein bioaccessibility and cellular uptake in bread were significantly different among the products, with wheat/HCS, 50/50 bread demonstrating the highest bioaccessibility and cell uptake of lutein (Figure 2A). The bioaccessibility of lutein for wheat/HCS, 50/50 bread was 79.7% and 59.7% without and with cleanup, respectively. Similarly, the wheat/HCS, 75/25 (55.3% and 29.4%) and wheat/HCS, 85/15 (47.2% and 32.7%), and finally, the wheat control bread (41.9% and 33.9%) had a lower bioaccessibility following digesta clean-up. The lutein bioaccessibility of flatbread made from einkorn wheat or einkorn fortified with lutein has a range of 23–54%, which is subject to lutein concentration [22]. The bioaccessibility of the lutein-fortified cupcake was in the range of 30–80% of the initial lutein content and subject to the fortification level (1–12 mg lutein per serving) [35]. These studies indicate that the fortification of staple foods could help diminish the risk of degenerative processes associated with lutein deficiency in humans. The current results are within the range of the bioaccessibility of lutein reported in fortified cupcakes and flatbreads. The current study also showed that lutein in baked products was bioaccessible and could potentially be available for intestinal absorption and colon fermentation. The apical uptake of lutein in wheat/HCS, 50/50, and wheat/HCS, 75/25 breads was 9.2 and 4.3%, respectively (Figure 2A). This indicates that a small portion of lutein could be absorbed in the small intestine. Significant differences in lutein uptake by Caco-2 cells between wheat/HCS, 50/50 bread, and wheat/HCS, 75/25 were also observed. The absence of lutein detected in Caco-2 cell harvest for the control and wheat/HCS, 85/15 breads could possibly be due to the initial low concentration of lutein in these breads. The higher HCS proportion in the bread formulation resulted in higher bioaccessibility and cell uptake of lutein.

Small concentrations of zeaxanthin were found in the bread digesta ranging from 0.06 to 0.09 µg/g and 0.03 to 0.06 µg/g for the non-cleaned-up and cleaned-up products, respectively (Table 3). No zeaxanthin was detected in the Caco-2 cell harvest. The bioaccessibility of zeaxanthin was in the range of 27.4–61.1% and 17.7–32.2% for bread products without and with cleanup, respectively, with the control bread demonstrating the highest bioaccessibility (Figure 2B). As expected, due to the initial small concentration of zeaxanthin in bread products, zeaxanthin uptake was not observed since both wheat and HCS are not sources of zeaxanthin. The zeaxanthin bioaccessibility of sweet corn (54%) and red pepper (48%) as non-green vegetables has been found to be higher than that of green vegetables, e.g., spinach (7%) and lettuce (6%) [36]. These results suggest that the food matrix and final digesta composition affect the micellarization of carotenoids and, eventually, their potential absorption by enterocytes and uptake by organs.

Reasonable concentrations of free ferulic acid were found in the control bread (5.8 µg/g) and composite bread (6.5–6.8 µg/g) for the non-cleaned-up digesta. In comparison, 5.1 µg/g and 5.3–6.8 µg/g of ferulic acid for cleaned-up digesta was observed in the control and composite bread, respectively (Table 3). The bioaccessibility of ferulic acid was fairly high, ranging from 98 to 103% and 81 to 102% for bread products without and with cleanup, respectively (Figure 2C). These results suggest that a small portion of bound ferulic acid was released during in vitro digestion. In another study, the contents of ferulic acid in wheat bread (buffer extracts) were potentially bioaccessible (in vitro digestion extracts) and potentially bioavailable (in vitro absorption extracts) and have been found to be 13.8, 12.3, and 5.4 µg/g, respectively [37]. This corresponds to a bioaccessibility in ferulic acid of approximately 89%. In the same study, the fortification of bread with green coffee beans at levels of 1–5% significantly improved the content of phenolic compounds and antioxidant activity, particularly at 5% enrichment. The bioaccessibility of ferulic acid seems to be relatively high and dependent on the food matrix and digesta composition, which was also confirmed in the current study. In a separate study, the bioaccessibility of ferulic acid from aleurone-enriched bread was found to be significantly higher (40.7%) than that of commercial wholegrain bread (13.1%) [38]. Furthermore, the same study showed that ferulic acid had a higher bioaccessibility than *p*-coumaric acid (29.5% & 10.1%) but was lower than sinapic (79.5% & 32.2%) and caffeic (83.3% & 19.2%) acids, respectively [38]. Despite a high bioaccessibility, ferulic acid had a very low uptake by Caco-2 cells at the level of 0.2% (Figure 2C). In Swieca’s study [37], the bioavailability potential was based on a dialysis method, and therefore, a comparison could not be made with the results of the current study. In another study, the bioavailability of ferulic acid in healthy individuals fed with the commercial bread (wholegrain versus aleurone-enriched) has been reported to be insignificantly different when the individuals were fed with the same amount of ferulic acid (87 mg) but with different bread amounts (94 versus 190 g). However, when the same amount of bread was given (94 g) and the total ferulic acid intake was different (87 versus 43 mg), the aleurone-enriched bread was significantly better absorbed and excreted than the control bread [39]. In another study, the absorption of ferulic and sinapic acids in the small intestine was limited and rapid, possibly coming from only the soluble fraction [40]. In contrast, the bound fraction proceeds to the colon in humans fed high-bran wheat breakfast cereal [40]. In addition, most of the ferulic acid detected in urine and plasma is present as conjugates of feruloylglycine and/or glucuronides. This could partly explain the low content of ferulic acid detected in the Caco-2 cell harvest in the current study.

## 4. Conclusions

The ability of carotenoids and polyphenols to modulate free radicals and inflammation in the body has been associated with the promotion of the health of eyes, brain, and cardiovascular system. Since humans are unable to synthesize these bioactive components and daily intake remains low worldwide, the consumption of carotenoid- and polyphenolic-fortified foods is an important component to ensure human health. The current study used HCS as a source of bioactive peptides, minerals, vitamins, carotenoids, and polyphenols to replace a portion of wheat flour in bread recipes and to make muffins from blends of HCS and corn with the intention of boosting the daily intake of carotenoids and phenolic acids. As such, the study used a combination of the simulated gastrointestinal digestive system and Caco-2 cells to estimate the potential absorption of lutein, zeaxanthin, and ferulic acid in the small intestine. Since the intestinal digests (as is) damaged the Caco-2 monolayer due to the high concentration of pancreatic enzymes and bile salts used in the simulated gastrointestinal digestion under the fed state conditions, it was necessary to clean up the digesta and remove these enzymes. The cleanup of intestinal digests with the C18 SampliQ column, along with the AEBSF inhibitor, was the most effective treatment, as the monolayer cells remained intact over the entire incubation period (4 h). Although cleanup treatment reduced lutein, zeaxanthin, and ferulic acid uptake by the monolayer, this step is not physiologically representative of in vivo conditions, and the actual uptake would be expected to be higher. Thus, the use of the in vitro co-culture of Caco-2 cells and HT29-MTX cells (a human mucus-producing cell line to protect Caco-2 cells) might represent a more physiological and realistic approach to in vivo conditions and may warrant future investigation. Lutein, zeaxanthin, and ferulic acid were bioaccessible from muffin and bread products and showed high bioaccessibility and potential absorption in the small intestine. Since the cellular uptake of these bioactive components was low, especially for bread products, it would be useful to enrich muffins and bread with higher amounts of these bioactive compounds. Moreover, these products could still boost the daily consumption of lutein, zeaxanthin, and ferulic acid, allowing for a small portion to be absorbed in the small intestine. In addition, the present study supports the development of functional foods that are rich in lutein, zeaxanthin, and/or ferulic acid through the enrichment and/or blending of multigrain flours, and especially the fortification of staple foods, to help diminish the risk factors of degenerative and chronic diseases in humans.

## Figures and Tables

**Figure 1 foods-12-01307-f001:**
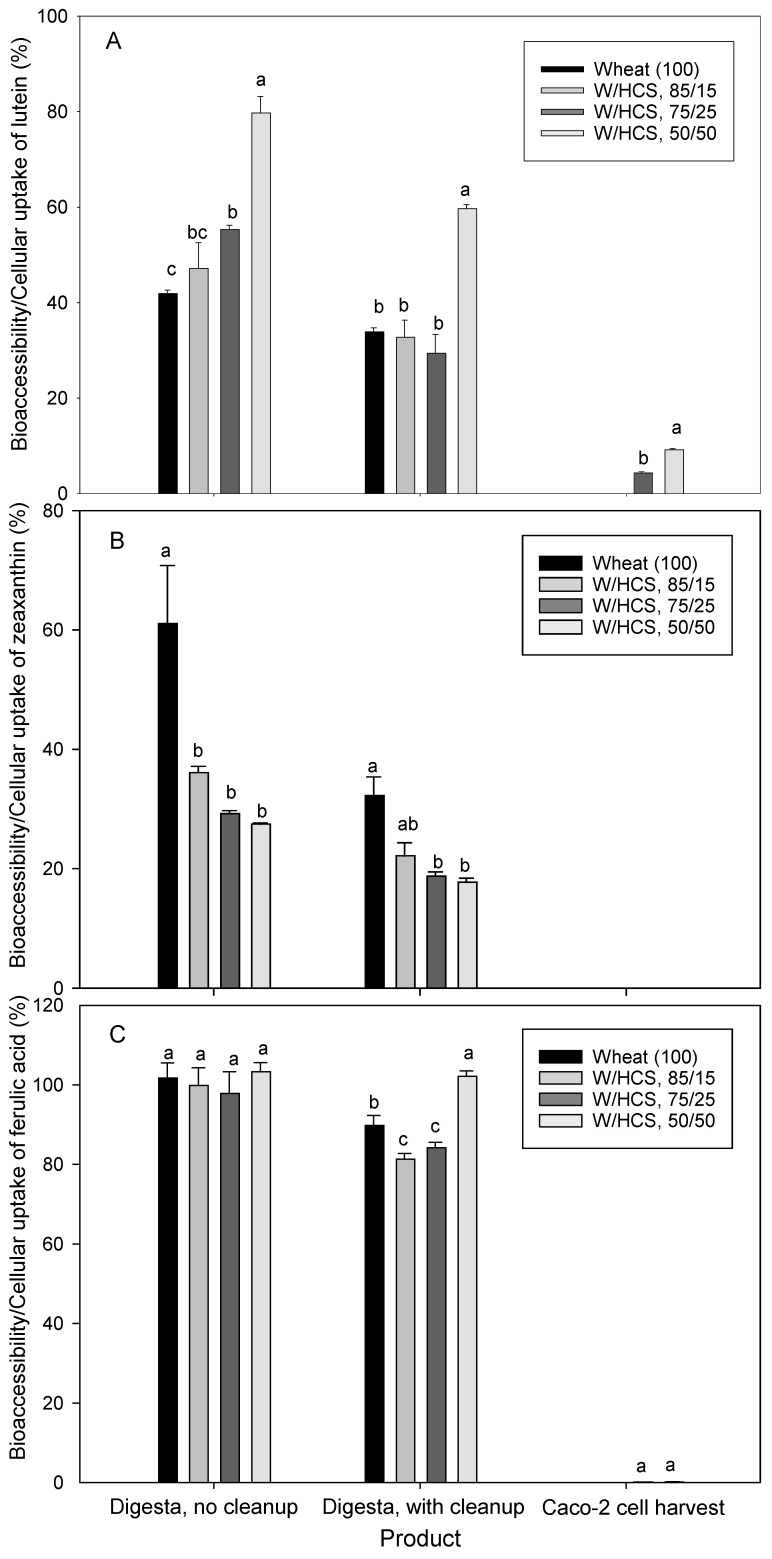
In vitro bioaccessibility of lutein from muffin intestinal digests without cleanup (**A**) and with cleanup (**B**) and its Caco-2 cellular uptake (**C**). Bars with the same letter are not significantly different at *p* < 0.05).

**Figure 2 foods-12-01307-f002:**
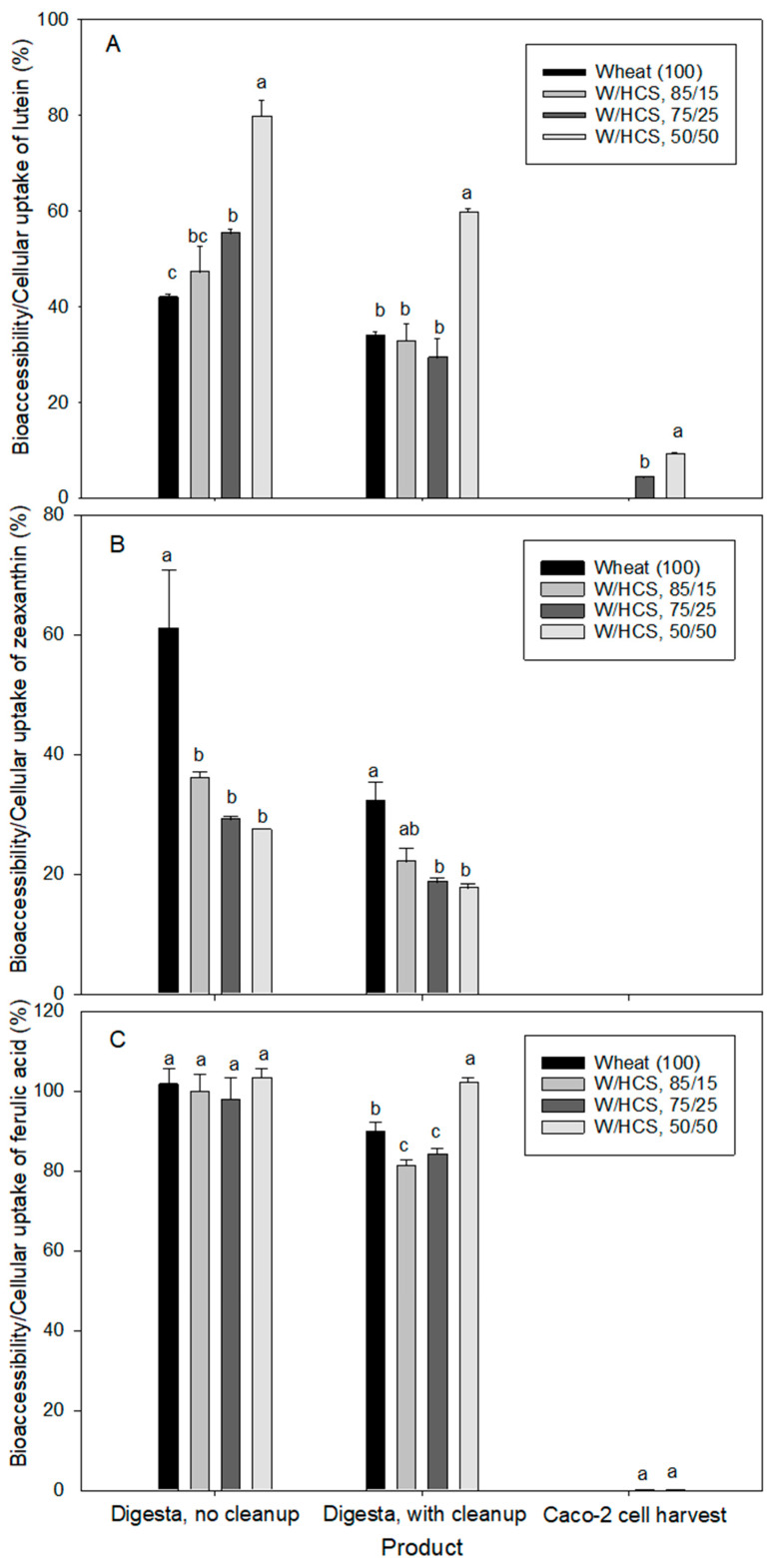
In vitro bioaccessibility of lutein from bread intestinal digests without cleanup (**A**) and with cleanup (**B**) and its Caco-2 cellular uptake (**C**). Bars with the same letter are not significantly different at *p* < 0.05).

**Table 1 foods-12-01307-t001:** Effect of various cleanup treatments of digesta on survival of caco-2 cells and status of lutein and ferulic acid following the simulated gastrointestinal digestion and Caco-2 cell incubation.

Treatment of Digesta	Caco-2 Cells after 4 h Incubation Time ^x^	Lutein ^y^		Ferulic Acid ^y^		Conclusion ^z^
		Digesta	Caco-2 Cells	Digesta	Caco-2 Cells	
As is (untreated)	MD	D	NM	D	NM	NA
Acid (pH 2.0)	MI	ND	ND	D	D	NA
Cooling in ice box	MD	NM	NM	NM	NM	NA
Liquid nitrogen	MD	NM	NM	NM	NM	NA
Dialysis (Amicon)	MI	ND	ND	D	D	NA
Amberlite XAD7	MD	D	NM	D	NM	NA
Amberlite IR120	MD	NM	NM	D	NM	NA
Oasis HLP	MD	D	NM	D	NM	NA
Alumina-N	MI	D	D	D	D	A
C18 SampliqQ	MI	D	D	D	D	A
AEBSF enz Inhib	MD	NM	NM	NM	NM	NA
PMSF enz Inhib	MD	NM	NM	NM	NM	NA
C18+AEBSF	MI	D	D	D	D	A
C18+PMSF	MI	D	D	D	D	A

^x^ Scale of 2 levels: monolayer intact (MI); monolayer detached (MD); ^y^ scale of 2 levels: detected (D); not detected (ND). NM stands for not measured; ^z^ applicable (A), it produces promising results and can be implemented; not applicable (NA), causes damage to cells and/or lutein/ferulic acid content not detected.

**Table 2 foods-12-01307-t002:** Actual concentration of lutein, zeaxanthin, and ferulic acid in muffins, their digesta without and with cleanup, and Caco-2 cell harvest (µg/g, as is basis, Mean ± SD) ^x^.

Products	Lutein	Zeaxanthin	Ferulic Acid
Muffins			
HCS (100%)	1.15 ± 0.057 ^b^	0.18 ± 0.042 ^c^	2.02 ± 0.191 ^a^
HCS/C (1:1, *w*/*w*)	1.22 ± 0.014 ^b^	1.10 ± 0.021 ^b^	1.29 ± 0.113 ^b^
HCS/C (1:2, *w*/*w*)	1.41 ± 0.021 ^a^	1.57 ± 0.021 ^a^	1.18 ± 0.042 ^b^
Intestinal digests (as is, no cleanup)			
HCS (100%)	1.09 ± 0.007 ^b^	0.12 ± 0.035 ^b^	2.10 ± 0.014 ^b^
HCS/C (1:1, *w*/*w*)	1.13 ± 0.014 ^b^	0.87 ± 0.049 ^a^	2.49 ± 0.106 ^a^
HCS/C (1:2, *w*/*w*)	1.29 ± 0.028 ^a^	1.05 ± 0.106 ^a^	2.70 ± 0.014 ^a^
Intestinal digests (after cleanup)			
HCS (100%)	0.94 ± 0.035 ^a^	0.09 ± 0.007 ^b^	1.07 ± 0.071 ^b^
HCS/C (1:1, *w*/*w*)	1.05 ± 0.085 ^a^	0.49 ± 0.042 ^a^	1.14 ± 0.007 ^b^
HCS/C (1:2, *w*/*w*)	1.16 ± 0.007 ^a^	0.60 ± 0.049 ^a^	1.57 ± 0.035 ^a^
Caco-2 cell harvest			
HCS (100%)	0.08 ± 0.007 ^a^	0.02 ± 0.002 ^a^	0.011 ± 0.001 ^b^
HCS/C (1:1, *w*/*w*)	0.11 ± 0.014 ^a^	0.07 ± 0.007 ^a^	0.012 ± 0.001 ^b^
HCS/C (1:2, *w*/*w*)	0.12 ± 0.024 ^a^	0.07 ± 0.007 ^a^	0.021 ± 0.002 ^a^

^x^ For each group, means in a column followed by a different superscript letter are significantly different at *p* < 0.05.

**Table 3 foods-12-01307-t003:** Actual concentration of lutein, zeaxanthin, and ferulic acid in breads, their digesta without and with cleanup, and Caco-2 cell harvest (µg/g, as is basis, Mean ± SD) ^x^.

Products	Lutein	Zeaxanthin	Ferulic Acid
Breads			
Wheat (100%)	0.31 ± 0.028 ^b^	0.10 ± 0.014 ^d^	5.67 ± 0.311 ^b^
Wheat/HCS (85/15, *w*/*w*)	0.52 ± 0.014 ^a^	0.18 ± 0.014 ^c^	6.55 ± 0.226 ^a^
Wheat/HCS (75/25, *w*/*w*)	0.62 ± 0.035 ^a^	0.24 ± 0.028 ^b^	6.61 ± 0.226 ^a^
Wheat/HCS (50/50, *w*/*w*)	0.70 ± 0.049 ^a^	0.31 ± 0.028 ^a^	6.62 ± 0.156 ^a^
Intestinal digests (as is, no cleanup)			
Wheat (100%)	0.13 ± 0.014 ^c^	0.06 ± 0.007 ^a^	5.76 ± 0.099 ^b^
Wheat/HCS (85/15, *w*/*w*)	0.25 ± 0.021 ^bc^	0.07 ± 0.007 ^a^	6.53 ± 0.071 ^a^
Wheat/HCS (75/25, *w*/*w*)	0.34 ± 0.014 ^b^	0.07 ± 0.007 ^a^	6.46 ± 0.141 ^a^
Wheat/HCS (50/50, *w*/*w*)	0.56 ± 0.064 ^a^	0.09 ± 0.007 ^a^	6.83 ± 0.007 ^a^
Intestinal digests (after cleanup)			
Wheat (100%)	0.11 ± 0.007 ^b^	0.03 ± 0.001 ^c^	5.09 ± 0.134 ^d^
Wheat/HCS (85/15, *w*/*w*)	0.17 ± 0.014 ^b^	0.04 ± 0.007 ^bc^	5.33 ± 0.092 ^c^
Wheat/HCS (75/25, *w*/*w*)	0.18 ± 0.014 ^b^	0.05 ± 0.007 ^ab^	5.56 ± 0.099 ^b^
Wheat/HCS (50/50, *w*/*w*)	0.42 ± 0.035 ^a^	0.06 ± 0.007 ^a^	6.76 ± 0.071 ^a^
Caco-2 cell harvest			
Wheat (100%)	nd	nd	nd
Wheat/HCS (85/15, *w*/*w*)	nd	nd	nd
Wheat/HCS (75/25, *w*/*w*)	0.02 ± 0.002 ^b^	nd	0.014 ± 0.001 ^a^
Wheat/HCS (50/50, *w*/*w*)	0.06 ± 0.005 ^a^	nd	0.016 ± 0.001 ^a^

^x^ For each group, means in a column followed by a different superscript letter are significantly different at *p* < 0.05.

## Data Availability

The data presented in this study are available in the article.

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
