# Peer review of "Bioaccessibility and Cellular Uptake of Lutein, Zeaxanthin and Ferulic Acid from Muffins and Breads Made from Hairless Canary Seed, Wheat and Corn Blends"

_foods, 2023, doi:10.3390/foods12061307_

Round 1
Reviewer 1 Report
The paper entitled "Bioaccessibility and cellular uptake of lutein, zeaxanthin and ferulic acid from muffins and breads made from hairless canary seed, wheat and corn blends" represents a high-quality study with a lot of novelty details in the experimental set-up.
Although I think that this paper is appropriate for the Foods, some improvements need to be added to the paper. These are my:
- simplicity of the Abstract is required
- Adding the state-of-the-art selected digestion method must be added. Please, compare with other mentioned digestion system(s).
- At the end of the Introduction add the aim and frame of this study.
- Although you mentioned previous references, add some details about the muffin formulas and baking process. The same comment for bread.
- Check the way for the presentation of the Reference list (I think that usually are smaller letters, and minimal line spacing)
Author Response
The paper entitled "Bioaccessibility and cellular uptake of lutein, zeaxanthin and ferulic acid from muffins and breads made from hairless canary seed, wheat and corn blends" represents a high-quality study with a lot of novelty details in the experimental set-up.
Although I think that this paper is appropriate for the Foods, some improvements need to be added to the paper. These are my:
- simplicity of the Abstract is required
The abstract was revised.
- Adding the state-of-the-art selected digestion method must be added. Please, compare with other mentioned digestion system(s).
This information is highlighted in the introduction “To better understand what happens to food after ingestion, foods or meals are subjected to in vitro digestion (salvia, gastric and intestinal) and conditions that simulate the human digestive system and render the food nutrients accessible. Coupled with an in vitro Caco-2 cell culture model, the uptake and transport (e.g. in vitro bioavailability) of nutrients and bioactive compounds [18, 19] can be evaluated. To date, several models exist, including the dynamic gastrointestinal model (e.g. TIM™), which is a sophisticated and expensive system, whereby many parameters of the human digestive system are simulated, while the Caco-2 cell model is widely used to assess nutrient uptake and bioavailability in vitro. ”.
- At the end of the Introduction add the aim and frame of this study.
The aim of the study was rephrased for clarity.
- Although you mentioned previous references, add some details about the muffin formulas and baking process. The same comment for bread.
Requested information was added.
- Check the way for the presentation of the Reference list (I think that usually are smaller letters, and minimal line spacing)
Line space for the reference section is 1, which is the minimal. A line space between each reference is to facilitate the revision, but has been deleted.

Reviewer 2 Report
The authors investigated bioaccessibility and cellular uptake of these components from muffins and breads made from blends of hairless canary seed (HCS), wheat and corn using simulated gastrointestinal digestion combined with Caco-2 cell model. The overall design of this work is reasonable. The study also has some implications, but there are more problems with the abstract and conclusions, methodology, figures and analysis of the results, especially the data processing part.
1. In abstract, please summarize the research background and strategic advantages in one sentence at the beginning. The end is too simple. What kind of domain value has been achieved? Research significance?
2. The "introduction" part introduces the background, nutritional value and some research progress of lutein, zeaxanthin, ferulic acid and Caco-2 cell model. First of all, the logic of the text before and after is not strong, please elaborate on the current research gaps in this field and highlight the research value of your research indicators. Second, the author cited a large number of literatures, but only a simple summary, lack of data summary. The authors are advised to continue to illustrate the strong prospects of the current research. It is hoped that the author will revise the preface carefully, highlight the research theme and significance, and add specific data to assist interpretation (4-5 paragraphs).
3. “2.3. Preparation of bread and muffin products”. Even if relevant literature is cited, the bread ingredients and baking conditions need to be clarified in detail in this section. This is related to the accuracy of the experiment, because the ingredients, baking temperature, time and other conditions will have a certain impact on the specific composition of the final sample.
4. “2.3. Preparation of bread and muffin products”. How is the ratio of wheat to HCS blends determined? Through pre-experiment or other literature? I can't see the regularity of proportion, equal difference or equal ratio?
5. “2.4. In vitro simulation of gastrointestinal digestion”. There is something wrong with the temperature symbol. Please check the full text carefully and modify it one by one. “℃”.
6. “We” should be avoided in the manuscript, and “the author” can be used instead.
7. “2.5. Cleanup treatments”. The description of the experimental method is too cumbersome, so it is suggested to describe it in points or deal with it in sections.
8. “2.6. Cellular uptake of lutein, zeaxanthin and ferulic acid using Caco-2 cell model”. How to maintain the condition of “maintained at 37 oC in a humidified atmosphere with 95% air and 5% CO2 (v/v)”? Please clarify.
9. Please determine whether there is a blank cell between the number and the unit, such as “using 0.2 μm GHP”and“(10 cm × 2.1mm × 1.8μ) in the section 2.7.
10. Generally, the specific data and error need to be consistent with valid figures, please check Table 2 and Table 3.
11. It is hoped that the bar chart can express the content more clearly. Please check whether the content in Figure 1 and 2 C (Caco-2 cell haevest) is correct. In addition, what does the legend show in all data graphs? Is it a drawing error? This is not a mistake that should occur. In addition, there are problems with the significant differences in the figure, which will inevitably affect the accuracy of the results and the objectivity of the analysis.
12. There are too many words in the conclusion part, and the elaboration of the non-experimental part is unnecessary. It is hoped that it can be deleted, please refine the experimental steps and summarize the conclusion. It helps readers to understand the aim of this work. In addition, the conclusion section recommends adding specific data to illustrate.
13. The research content of this manuscript is interesting, but there are more or less problems in the logic of the introduction, the choice of key steps in the methods, the processing of data, and the summary of conclusion. It is suggested that the author read more other articles of this journal in order to better write the manuscript.

Author Response
In abstract, please summarize the research background and strategic advantages in one sentence at the beginning. The end is too simple. What kind of domain value has been achieved? Research significance?
The abstract was revised.
The "introduction" part introduces the background, nutritional value and some research progress of lutein, zeaxanthin, ferulic acid and Caco-2 cell model. First of all, the logic of the text before and after is not strong, please elaborate on the current research gaps in this field and highlight the research value of your research indicators. Second, the author cited a large number of literatures, but only a simple summary, lack of data summary. The authors are advised to continue to illustrate the strong prospects of the current research. It is hoped that the author will revise the preface carefully, highlight the research theme and significance, and add specific data to assist interpretation (4-5 paragraphs).
The introduction was revised for more clarity
“2.3. Preparation of bread and muffin products”. Even if relevant literature is cited, the bread ingredients and baking conditions need to be clarified in detail in this section. This is related to the accuracy of the experiment, because the ingredients, baking temperature, time and other conditions will have a certain impact on the specific composition of the final sample.
Information was added.
- “2.3. Preparation of bread and muffin products”. How is the ratio of wheat to HCS blends determined? Through pre-experiment or other literature? I can't see the regularity of proportion, equal difference or equal ratio?
The blend ratios are based on the HCS flour replacement, and were stated in the material and method section 2.3.
“2.4. In vitro simulation of gastrointestinal digestion”. There is something wrong with the temperature symbol. Please check the full text carefully and modify it one by one. “℃”.
The symbol was replaced with the proper one.
“We” should be avoided in the manuscript, and “the author” can be used instead.
Sentence was revised and “we’ was deleted.
“2.5. Cleanup treatments”. The description of the experimental method is too cumbersome, so it is suggested to describe it in points or deal with it in sections.
Section was revised and rephrased for clarity.
“2.6. Cellular uptake of lutein, zeaxanthin and ferulic acid using Caco-2 cell model”. How to maintain the condition of “maintained at 37 oC in a humidified atmosphere with 95% air and 5% CO2 (v/v)”? Please clarify.
This was done in a humidity controlled 5% CO2 incubator. Sentence was rephrased to reflect this information.
Please determine whether there is a blank cell between the number and the unit, such as “using 0.2 μm GHP”and“(10 cm × 2.1mm × 1.8μ) in the section 2.7.
Blank space has been deleted.
Generally, the specific data and error need to be consistent with valid figures, please check Table 2 and Table 3.
The manuscript has gone through extensive English Editing.
It is hoped that the bar chart can express the content more clearly. Please check whether the content in Figure 1 and 2 C (Caco-2 cell haevest) is correct. In addition, what does the legend show in all data graphs? Is it a drawing error? This is not a mistake that should occur. In addition, there are problems with the significant differences in the figure, which will inevitably affect the accuracy of the results and the objectivity of the analysis.
The figures were generated using SigmaPlot statistical and data analysis software. The program generate the bar errors as well as the data legend.
The file submitted has no such issues at our ends. We should be able to address this problem during the galley proof and in collaboration with the print department.
There are too many words in the conclusion part, and the elaboration of the non-experimental part is unnecessary. It is hoped that it can be deleted, please refine the experimental steps and summarize the conclusion. It helps readers to understand the aim of this work. In addition, the conclusion section recommends adding specific data to illustrate.
Conclusion was revised and unnecessary information was deleted.
The research content of this manuscript is interesting, but there are more or less problems in the logic of the introduction, the choice of key steps in the methods, the processing of data, and the summary of conclusion. It is suggested that the author read more other articles of this journal in order to better write the manuscript.
The manuscript has gone through extensive English language and grammar editing.

Reviewer 3 Report
The topic of this paper is interesting for the scientific community and the paper is well-written. However, it would be convenient to review some aspects. Please see comments below.
GENERAL:
· Even though the article is very well written and easy to follow, the text has many grammatical errors (double spaces, capital letters...), please check all the text.
INTRODUCTION
· “The role of lutein in the health of eyes” --> Also indicates the effects of zeaxanthin on human health.
· Spell “GRAS” out the first time.
· (e.g. salvia, gastric, and intestinal) --> (saliva, gastric and intestinal)
· (e.g. TIM)™ --> (e.g. TIM ™)
· “This is due to the ability of Caco-2 cells” --> I think it is not necessary to include this sentence, it could be deleted. Caco-2 cells are quite familiar to the vast majority of scientists.
· “The Caco-2 cell model was developed and used in assessing the relative bioavailability of carotenoids from various baby food meals” --> Are there no articles in the literature that evaluate the uptake of carotenoids from matrices similar to those studied in this article using Caco-2 cells? If there were, it would be better to discuss those articles than the one on baby food.
MATERIALS AND METHODS
· The acronym “MtBE” is not necessary, it is only used once.
· Have the carotenoids been analyzed to detect which are the main isomers? Were the evaluated carotenoids cis or trans? This is something important to mention.
· “75 U/mL” --> Has the actual activity of each enzyme used in the in vitro digestion model been determined?
· It seems that only three replicates were used in the in vitro digestion protocol --> Indicate how many replicates were used. If only three were used, increasing the number of replicates should be considered for future studies.
· “37 o﮿C. ” --> 37 ﮿C.
· “The carotenoids and phenolic acids were present in fraction 1 only in all columns and thus it was further investigated.” --> The carotenoids and phenolic acids were present in all columns only in fraction 1, and thus it was further investigated.
· “The final cleanup process of digesta was based on the use of C18 SPE SampliQ column.” --> Please, rewrite the sentence, not sure if it is a final cleanup step or if it is the method selected as the best one.
· “then extracted with 0.3 mL of water-saturated butanol for carotenoid analysis” --> Was the extraction carried out until colorless?
· (15 cm x 4.6 mm x 3μ) --> (15 cm x 4.6 mm x 3μm)
· “The bioaccessibility is defined as the amount […] The data are also presented in absolute amounts for comparison purposes”. --> Move this paragraph to sections 2.4 and 2.6.
RESULTS AND DISCUSSION
· Results from other similar studies should be discussed in section 3.1.
· It would be appropriate to include a figure of the three levels of damage to the Caco-2 cells.
· “As a result, the pH treatment is considered an inappropriate method for assessing lutein bioavailability in vitro […] of invulnerable nutrients such as minerals.” --> Please, rewrite the sentences. Repeated and irrelevant information (Is heat treatment used in this study during the pH change? If not, remove the information on the effect of heat treatment on lutein as it is confusing)
· TABLE 1: Why is Acid (pH 2) considered not applicable (NA)? NA is stated to mean cause damage to Caco-2 cells. However, with Acid (pH 2) the monolayer is intact. It seems that NA considers damage to Caco-2 cells as well as carotenoids and ferulic acid content. If so, please modify the information about the meaning of NA.
· Why is zeaxanthin not considered in Table 1?
· I think it would be convenient to include new sections on carotenoids and ferulic acid concentrations in muffins and breads.
· “Other carotenoids are also present in HCS and corn with a total carotenoid content of 7.6 and 12.9 μg/g, respectively.” --> Indicate the reference if this result is from a previous study. If not: “Other carotenoids WERE also…”
· Indicate if the samples for carotenoid analysis have been saponified. Have all the isomers of each carotenoid been quantified together? For the concentrations indicated in Table 2, have carotenoid esters been taken into account?
· The concentration of soluble ferulic acid in muffins […] made from HCS and corn”. --> Please, rewrite the sentence. Repeated information.
· “The high fat content in muffins improve stability of lutein in the micelles, and thus its bioaccessibility [22]. The study has also…” --> The high fat content in muffins HAVE BEEN SHOWN TO improve stability […]. THIS STUDY has…
· Figure 1 and 2 --> THE COLORS THAT REPRESENT EACH OF THE FOOD PRODUCTS ARE NOT VISIBLE!
· In another study, the bioaccessibility of ferulic acid from aleurone-enriched bread (40.7%) […]. respectively [38].” --> Please, check the sentence.
· “Since the cellular uptake of these bioactive components, was low especially for bread products, it would be useful to enrich muffins and breads with higher amounts of bioactive compounds.” --> It would be appropriate to indicate that the cellular uptake can be a saturable process, therefore, the increase in the concentration of bioactive compounds does not always translate into an increase in the amount uptaken, as indicated in this and other articles: (1)P. Mapelli-Brahm et al., Mol. Nutr. Food Res., in press, doi:10.1002/mnfr.201800703.
Author Response
Even though the article is very well written and easy to follow, the text has many grammatical errors (double spaces, capital letters...), please check all the text.
As mentioned above, the manuscript has gone through extensive English language editing.
INTRODUCTION
- “The role of lutein in the health of eyes” --> Also indicates the effects of zeaxanthin on human health.
Information was added.
- Spell “GRAS” out the first time.
Information was added.
- (e.g. salvia, gastric, and intestinal) --> (saliva, gastric and intestinal)
Suggested change was applied.
- (e.g. TIM)™ --> (e.g. TIM ™)
Suggested change was applied.
- “This is due to the ability of Caco-2 cells” --> I think it is not necessary to include this sentence, it could be deleted. Caco-2 cells are quite familiar to the vast majority of scientists.
Sentence was deleted.
- “The Caco-2 cell model was developed and used in assessing the relative bioavailability of carotenoids from various baby food meals” --> Are there no articles in the literature that evaluate the uptake of carotenoids from matrices similar to those studied in this article using Caco-2 cells? If there were, it would be better to discuss those articles than the one on baby food.
There is a limited information regarding the use of Caco-2 cells to evaluate the uptake of carotenoids from cereal products. There are some published papers on carotenoids but these are related to the bioavailability (using transwells) and not uptake. A number of papers looked at iron bioavailability in breads. Carotenoid bioavailability papers relate more to the plant foods (carrots, spinach etc).The baby food study cited in the introduction has an extensive analytical determination of various carotenoids. Our previous work, which has been conducted on baked products is also cited in the subsequent sentences.
MATERIALS AND METHODS
- The acronym “MtBE” is not necessary, it is only used once.
Abbreviation was deleted.
- Have the carotenoids been analyzed to detect which are the main isomers? Were the evaluated carotenoids cisor trans? This is something important to mention.
Extensive analysis of carotenoids was conducted in our citied reference [1] and mentioned in the introduction “The main carotenoids found in HCS comprised of lutein and its mono- and di-esters in cis and trans forms [1]. The amount of lutein presented in this study is the sum of all lutein isomers including trans-lutein and a possible co-eluted cis- lutein. Information was added.
- “75 U/mL” --> Has the actual activity of each enzyme used in the in vitro digestion model been determined?
Enzymes activity is provided by the supplier in the product specifications.
- It seems that only three replicates were used in the in vitro digestion protocol -->Indicate how many replicates were used. If only three were used, increasing the number of replicates should be considered for future studies.
Three replicates were completed for the in vitro digestion protocol. Each of these replicates were then divided again in triplicate for cell culture uptake.
- “37 o﮿C. ” --> 37 ﮿C.
Suggested change was applied.
- “The carotenoids and phenolic acids were present in fraction 1 only in all columns and thus it was further investigated.” --> The carotenoids and phenolic acids were present in all columns only in fraction 1, and thus it was further investigated.
Suggested change was applied.
- “The final cleanup process of digesta was based on the use of C18 SPE SampliQ column.” --> Please, rewrite the sentence, not sure if it is a final cleanup step or if it is the method selected as the best one.
Sentence was rephrased for clarity.
- “then extracted with 0.3 mL of water-saturated butanol for carotenoid analysis” --> Was the extraction carried out until colorless?
Extraction is replaced by re-dissolved for more clarity.
- (15 cm x 4.6 mm x 3μ)--> (15 cm x 4.6 mm x 3μm)
Suggested change was applied.
“The bioaccessibility is defined as the amount […] The data are also presented in absolute amounts for comparison purposes”. --> Move this paragraph to sections 2.4 and 2.6.
Information was moved to the suggested sections.
RESULTS AND DISCUSSION
- Results from other similar studies should be discussed in section 3.1.
This section highlight various treatments used to optimize the use of Caco2-cells as a biological tool to study the absorption of carotenoids and phenolic acids in the small intestine. There is limited data on this particular area. Some related studies was cited in the section (Ref 18, 27, 28 29, 30).
- It would be appropriate to include a figure of the three levels of damage to the Caco-2 cells.
Table 1 has been revised to better consolidate research findings related to method development and selection of the procedure best-suited for this project. The selection of method was dependent on (1) the method didn’t damage the Caco-2 monolayer and (2) lutein/ferulic acid content could be detected. To clarify this decision process, the table has been adjusted to only include results at the end of the 4 incubation period with Caco-2 cells, with the monolayer either intact (MI) or damaged (MD). We believe this descriptor more accurately reflects our selection criteria. To that extent, the intact monolayer (MI) was 100% confluent whereas monolayer damaged (MD) represents any damage to the monolayer rendering the cells unfit for subsequent use. Visual observations using a Vista Vision inverted cell culture microscope were made to assess the suitability of the monolayer following each experimental procedure.
- “As a result, the pH treatment is considered an inappropriate method for assessing lutein bioavailability in vitro […] of invulnerable nutrients such as minerals.” --> Please, rewrite the sentences. Repeated and irrelevant information (Is heat treatment used in this study during the pH change? If not, remove the information on the effect of heat treatment on lutein as it is confusing)
Irrelevant information was deleted.
- TABLE 1: Why is Acid (pH 2) considered not applicable (NA)? NA is stated to mean cause damage to Caco-2 cells. However, with Acid (pH 2) the monolayer is intact. It seems that NA considers damage to Caco-2 cells as well as carotenoids and ferulic acid content. If so, please modify the information about the meaning of NA.
Table legend was changed to read “Not Applicable (NA), causes damage to cells and/or lutein/ferulic acid content not detected”
- Why is zeaxanthin not considered in Table 1?
Table 1 shows the effect of various cleanup treatments of digesta on survival of caco-2 cells and status of lutein and ferulic acid. As both lutein and zeaxanthin are isomers and their structure are quite similar, we present lutein as a major carotenoid and ferulic acid as the major phenolic acid.
- I think it would be convenient to include new sections on carotenoids and ferulic acid concentrations in muffins and breads.
The study objective is to investigate the bioaccessibility of carotenoids and ferulic acid. The concentrations of these components in muffins and breads depended on the clean up process and was affected by the digestion process. Thus, it was discussed under both sections, treatments of digests and bioaccessibilty.
- “Other carotenoids are also present in HCS and corn with a total carotenoid content of 7.6 and 12.9 μg/g, respectively.” --> Indicate the reference if this result is from a previous study. If not: “Other carotenoids WERE also…”
Citation was inserted.
- Indicate if the samples for carotenoid analysis have been saponified. Have all the isomers of each carotenoid been quantified together? For the concentrations indicated in Table 2, have carotenoid esters been taken into account?
Carotenoid were extracted using water saturated butanol solution. No saponification was required for the analysis. Isomers of carotenoids, as well as carotenoid esters have been quantified together, please refer to reference #1.
- The concentration of soluble ferulic acid in muffins […] made from HCS and corn”. --> Please, rewrite the sentence. Repeated information.
Sentence was rephrased.
- “The high fat content in muffins improve stability of lutein in the micelles, and thus its bioaccessibility [22]. The study has also…” --> The high fat content in muffins HAVE BEEN SHOWN TO improve stability […]. THIS STUDY has…
Suggested changes were applied.
- Figure 1 and 2 --> THE COLORS THAT REPRESENT EACH OF THE FOOD PRODUCTS ARE NOT VISIBLE!
We believe this may be due to computer setup, as the colors chosen are quite distinguish. If this issue still not clear in the galley proof, it will be fixed.
- In another study, the bioaccessibility of ferulic acid from aleurone-enriched bread (40.7%) […]. respectively [38].” --> Please, check the sentence.
Sentence was rephrased.
- “Since the cellular uptake of these bioactive components, was low especially for bread products, it would be useful to enrich muffins and breads with higher amounts of bioactive compounds.” --> It would be appropriate to indicate that the cellular uptake can be a saturable process, therefore, the increase in the concentration of bioactive compounds does not always translate into an increase in the amount uptaken, as indicated in this and other articles: (1)P. Mapelli-Brahm et al., Mol. Nutr. Food Res., in press, doi:10.1002/mnfr.201800703.
The conclusion statement highlight the important of enriching the muffin and bread products with lutein, zeaxanthin and ferulic acid which can boost the daily consumption of these bioactive compounds in the body, not under the in vitro cellular uptake. The sentence was rephrased for clarity.

Round 2
Reviewer 2 Report
作者已经解决了我的担忧。